# Understanding the Relations between Surface Stress State and Microstructure Feature for Enhancing the Fatigue Performance of TC6 Titanium Alloy

**Song Shu** [1,2], **Xin Huang** [3], **Zonghui Cheng** [1], **Yizhou Shen** [1,3,*], **Zhaoru He** [3] **and Weilan Liu** [4]

[1]  State-Owned Machinery Factory in Wuhu, Wuhu 241007, China; shusong2021@126.com (S.S.); 13625535177@139.com (Z.C.)

[2]  Science and Technology on Plasma Dynamics Laboratory, Air Force Engineering University, Xi'an 710038, China

[3]  College of Material Science and Technology, Nanjing University of Aeronautics and Astronautics, Nanjing 211100, China; 981101hx@nuaa.edu.cn (X.H.); hezhaoru@nuaa.edu.cn (Z.H.)

[4]  Institute of Advanced Materials, Nanjing Tech University, Nanjing 210009, China; iamwlliu@njtech.edu.cn

*  Correspondence: shenyizhou@nuaa.edu.cn

**Abstract:** Fatigue performance has always been an important factor affecting the application of titanium alloy. The service life of TC6 titanium alloy is easily reduced under a continuously alternating load. Therefore, there is an urgent need for a new method to improve fatigue performance. Laser shock peening (LSP) is a widely proposed method to enhance the fatigue performance. Here, through experiments and finite element simulations, it was found that LSP can prolong the fatigue life of TC6 by improving the surface stress state. In strengthening processes, the generation of residual stress was mainly attributed to the change of microstructure, which could be reflected by the statistical results of grain sizes. The content of grains with a size under 0.8 μm reached 78%, and the microhardness value of treated TC6 was 18.7% higher than that of an untreated sample. In addition, the surface residual compressive stress was increased to −600 MPa at the depth of 1500 μm from the surface. On this basis, the fatigue life was prolonged to 135%, and the ultimate fracture macroscopic was also changed. With the treatment of LSP, the fatigue performance of TC6 is highly promoted. The strengthening mechanism of LSP was established with the aim of revealing the relationship between microstructure and stress state for enhancing the fatigue performance in whatever shapes.

**Keywords:** laser shock peening; surface stress state; fatigue performance; titanium alloy

## 1. Introduction

Due to its low density, high specific strength, good corrosion resistance and outstanding processing performance, titanium alloy is an excellent structural material in many engineering fields. TC6 titanium alloy is an α + β two-phase titanium alloy with excellent comprehensive mechanical properties [1–3]. This alloy with the advantages of high specific strength and good corrosion resistance, also has superior plasticity in any shapes and a high-temperature (at 400–450 °C) work capacity. TC6 is mainly used to manufacture key parts in aerospace fields, such as compressor discs and blades as well as aircraft bulkheads and fasteners [4]. Nevertheless, in the course of service, fractures are often triggered on the surface under the action of continuously alternating load. These fractures gradually expand and then reduce the service life. Therefore, a large number of studies have been performed to look for the surface strengthening method for TC6 titanium alloy.

Owing to the existence of tensile stress, material surface is easy to initiate the fatigue crack under repetitively loading conditions. Commonly, the surface strengthening methods mainly include ultrasonic surface mechanical attrition treatment (SMAT) and low-plasticity rolling. The SMAT method requires an ultrasonic set-up to vibrate spherical, which is available to induce a nano-sized crystalline layer. Nonetheless, the impact of balls leading to

formation of valleys and peaks would increase the surface roughness [5–7]. Another surface strengthening method (i.e., low-plasticity rolling) can obtain a deeper strengthened layer and lower surface roughness [8–10]. However, the rolling strengthening method has strict material requirements in the shape aspect. Therefore, traditional strengthening methods are not adequate to meet the increasing demands of modern metal components, and some new strengthening methods have been proposed to replace these traditional ones.

Laser shock peening (LSP) is an advanced surface strengthening method that addresses the shortcomings of traditional methods. The cost of the LSP machine is higher than that of shot peening or rolling. LSP is a non-contact method with the advantages of high controllability, high flexibility, and high efficiency. The mechanism of LSP is irradiating a metal surface with a short pulsed and high peak power density laser [11–13]. In response, the metal surface forms a high-pressure plasma wave, leading to the enhancement of surface mechanical properties. LSP can change the surface microstructure in a certain internal depth. This induces the surface to produce residual compressive stress, high-density dislocations, as well as surface nanometer grains [14–16]. Therefore, LSP can significantly improve the fatigue resistance at room temperature, wear resistance, and stress corrosion resistance of metal materials [17,18]. Due to its excellent potential, LSP has received extensive attention from the aerospace industries. Up until now, many scholars have studied the relationship between surface stress state and microstructure feature. Luo, et al. [19] explored the influence of LSP combined with SP on the microstructure and fatigue performance of Ti-6Al-4V titanium alloy. The research focused on the change of microstructure for enhancing the fatigue performance in titanium alloy.

The present work mainly focused on the relationship between refined grain regions and residual stress during LSP, so as to strengthen fatigue performance. With high loading stress, we investigated the influence of LSP on the fatigue life and fracture behavior of TC6 titanium alloy. Also, the variation of stress state during the period of fatigue fracture was further analyzed via the simulation method, and the experimental results were used to verify the corresponding mechanism analyses on the strengthening action.

## 2. Experiments and Simulations

### 2.1. Materials and Components

The chemical composition of TC6 titanium alloy is listed in Table 1. The size of the samples was 30 mm × 30 mm × 10 mm, and the 25 mm × 12 mm areas outlined by the red dashed line were strengthened by LSP, as shown in Figure 1a. The fatigue test regions are shown in Figure 1b. The samples were cut by machine, and the original surface roughness Ra was measured as better than 0.8 μm. After machining the samples, they were pretreated by grinding and polishing. The processed surfaces of the samples were cleaned with alcohol and dried with cold air to ensure uniform roughness. Finally, LSP was performed on the entire gauge length section within the red dashed frame.

**Table 1.** Chemical composition of TC6 titanium alloy (wt.%).

| Al | Mo | Cr | Fe | Si | Ti |
|------|------|------|------|------|------|
| 6.09 | 2.60 | 1.54 | 0.49 | 0.31 | Bal. |

### 2.2. Experimental Procedures

In the LSP process, YS100-R200A equipment (Xi'an Tyrida Optical Electric Technology Co., Ltd., Xi'an, China) was used to carry out the surface treatment, as exhibited in Figure 2a. Maximum energy output was of 10 J. The pulse width was 18–20 ns and the focus spot size was 2–5 mm. The position accuracy of the six-joint robot was ±0.2 mm, and it was controlled in the x–y direction. The spot diameter during LSP was 2.6 mm at the 50% overlapping rate. Figure 2b was the real circumstance of laser action on the TC6 titanium alloy plate. Figure 2c shows that the flowing water layer was selected as the

constraining layer. Also, 0.1 mm opaque black tape was chosen to captivate the energy as an absorbing layer for all samples.

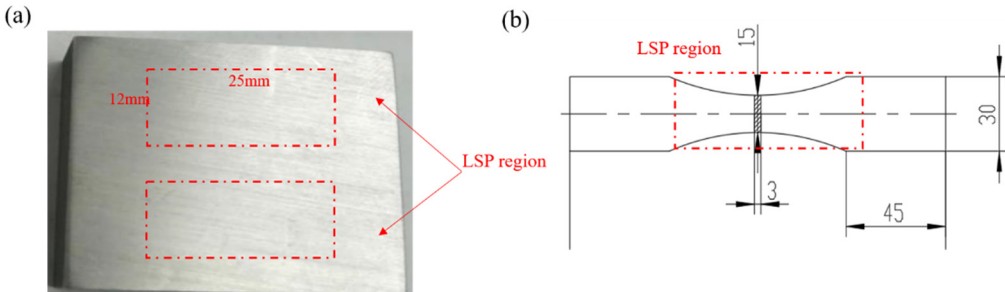

**Figure 1.** (**a**) TC6 titanium samples, (**b**) laser shock peening (LSP)-treated zone.

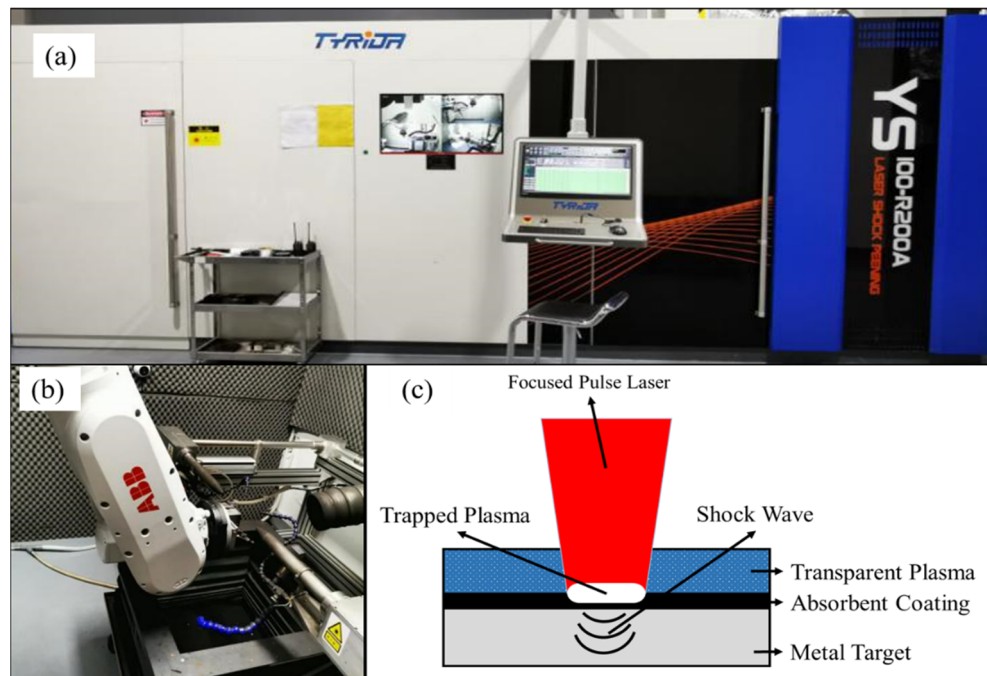

**Figure 2.** (**a**) LSP equipment, (**b**) LSP process, (**c**) schematic illustration of LSP treatment on sample surfaces.

When the peak pressure of the laser shock wave exceeds the Hugoniot elastic limit ($\sigma_{HEL}$) of the metal material, the fatigue performance of the metal can be effectively improved. The $\sigma_{HEL}$ of metallic materials is determined by Formula (1). Where $\sigma_y^{dyn}$ is the dynamic response yield strength at high strain rate (MPa) and $\lambda$ and $\alpha$ are Raman constants related to the material's elastic modulus E and Poisson's ratio. The wave pressure of the laser is estimated by Formula [20]:

$$\sigma_{HEL} = \frac{\lambda + 2\mu}{2\mu}\sigma_y^{dyn} \tag{1}$$

The elastic modulus E of TC6 is 123 GPa, the acoustic impedance Z is $2.73 \times 10^6$ g/(cm$^2$·s), and the dynamic yield strength is 1600 MPa. It is calculated that the minimum power density required for LSP of TC6 titanium alloy is 3.69 GW/cm$^2$.

In order to study the influence of the laser energy density and the number of impacts on the surface properties of TC6 titanium alloy [21,22], four groups of different processing

parameters of LSP treatment were selected to perform the strengthening treatment on these sample surfaces. The different LSP processing parameters are listed in Table 2.

**Table 2.** LSP processing parameters for TC6 titanium alloy samples.

| Number | Laser Power/J | Spot Diameter/mm | Overlap Rate | Pulse | Laser Fluence/GW/cm² |
|--------|---------------|------------------|--------------|-------|----------------------|
| Ti-1 | 4 | | | 1 | 3.77 |
| Ti-2 | 5 | | | 1 | 4.71 |
| Ti-3 | 6 | 2.6 | 50% | 1 | 5.65 |
| Ti-4 | 5 | | | 2 | 4.71 |

### 2.3. Measurement Apparatuses and Methods

A LEICA DMI 3000 M metallographic microscope (Leica Microsystems Ltd., Wetzlar, Germany) was used to observe the change of grain morphology. The surfaces roughness was tested by SRA-1 surface roughness tester (Shanghai shangguang optical instrument Co., Ltd., Shanghai, China). As for the residual stress, the equipment consisted of an LXRD residual stress tester (Proto Manufacturing Ltd., LaSalle, ON, Canada) and 8818 V-3 electrolytic polishing device (Shenzhen Kepuda Electromechanical Equipment Co., Ltd., Shenzhen, China). A proto-LXRD X-ray diffractometer (Proto Manufacturing Ltd., LaSalle, ON, Canada) can test the residual stress via the sin2φ method. The Cu-kα characteristic X-ray beam diameter was 2 mm, diffracted from a Ti plane was detected with a 2θ of 122°–162° [23] in 150 Hz. After electrolytic polishing, HX-1000TM/LCD equipment (Shanghai optical instrument factory, Shanghai, China) was used to test the microhardness under 200 g load for 15 s.

Titanium alloy is sensitive to stress concentration [24,25], and we pasted an electrical sheet in the holding section, as shown in Figure 3. In order to test fatigue performance, a QBG-100 high-frequency fatigue testing machine (Dalian Longrun Technology Co., Ltd., Dalian, China) was performed at room temperature under 150 Hz. The loading stress was from 500 MPa to 580 MPa. Under the action of continuously alternating load produced by the shaker, the fracture toughness test was carried out on the specimens to test the crack growth rate and the fracture threshold value of TC6 alloy. The stress ratio was 0.1, when the vibration frequency of the excitation system was equal to the natural frequency of the system itself. The system resonates, and the small excitation force generated is amplified and acted on the specimen during the material fatigue test.

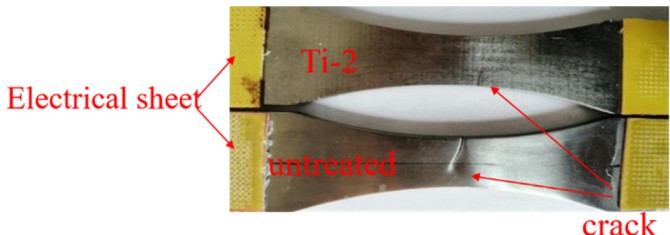

**Figure 3.** Appearance of fatigue fracture samples for untreated TC6 titanium alloy and LSP-treated Ti-2.

### 2.4. Finite Element Simulations

Sun, et al. [26] reported that multiple fatigue cracks were found at the corners of the main bearing frame and the wing wall panels of the entire aircraft. LSP, as an efficient surface strengthening method, can be used to reinforce the rounded area. In order to comprehend the effect of fatigue life on the frame beam structure of a fighter aircraft, the enhancement effect was observed by means of finite element method (FEM) simulations. Including mechanical and metallurgical changes, the process of LSP is transient. It needs finite element methods to analyze the dynamic changes of the material. ABAQUS/Explicit (Dassault Systemes SIMULIA, Providence, RI, USA) have been utilized to analyze the stress

state. In these models, the cases are solved in different linear and non-linear equations. The dynamic equation can be expressed as follows:

$$\int_{\Omega} \rho \ddot{u} \delta u_i d\Omega + \int_{\Omega} \vartheta \rho \dot{u} \delta u_i d\Omega = \int_{\Omega} f_i \delta u_i d\Omega + \int_{\tau} T_{\tau} \delta u_i d\tau - \int_{\Omega} \sigma_{ij} \delta D_{ij} d\Omega \tag{2}$$

where $\Omega$ means the global scope and $\tau$ means bounds, $\rho$ is mass density, $\ddot{u}$ and $\dot{u}$ nodal acceleration and velocity, respectively, $\vartheta$ the damping coefficient, $\delta u$ the virtual displacement, $f_i$ the body force density, $T_{\tau}$ the boundary force applied on the boundary, $\sigma_{ij}$ the Cauchy stress tensors, $D_{ij} = \frac{1}{2}(\delta u_{i,j} + \delta u_{j,i})$ is deformation rate tensors [27].

The LSP simulation is performed using ABAQUS software (version 6.14), according to aircraft beam structure. The 3D model is shown in Figure 4a. A general purpose brick element with reduced integration C3D8R is used for same shape function of C3D8, and C3D8R is a three-dimensional continuous displacement/stress of an eight-node element. In this model, the two planes of one end of the structural simulator were set as fixed constraints to avoid shock wave reflections, and the displacement U in the length direction of two planes of the other end was set as 0.16 mm. The meshing model was performed with the displacement U of 0.5 mm, consisting of 259556 C3D8R elements in Figure 4b. Results of FEM analysis are sensitive to the mesh density. It is easy to obtain accurate results with dense mashes, but at higher computational cost. To accurately capture mechanical effects induced by laser shock, it is necessary to have sufficient mesh density about the region, and also the region of the LSP area mesh needs to be adjusted. A Gaussian shape pressure pulse was used to simulate a shock wave produced by laser. For this pressure pulse, pressure rose sharply in the first few nanoseconds and then decreased gradually. Thus, the pressure load in relation to time was designed in a Gaussian wave shape to imitate reality.

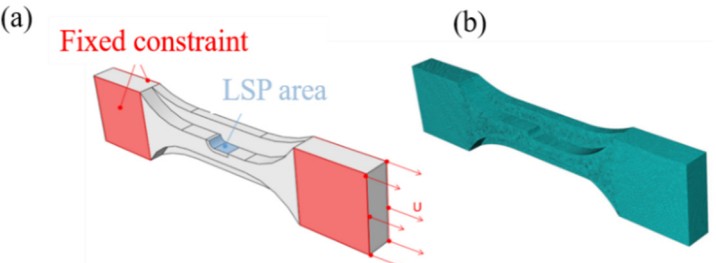

**Figure 4.** (**a**) Frame beam model of TC6 titanium alloy, (**b**) mesh model.

ABAQUS static results were imported into FE-SAFE software (version 2016) to calculate the fatigue life with the help of the Brown–Miller critical plane algorithm [28]. When the peak pressure of the laser shock wave exceeds the Hugoniot elastic limit ($\sigma_{HEL}$) of the metal material, the fatigue performance of the metal can be effectively improved, the fatigue performance of the metal can be effectively improved.

$$P = 0.01\sqrt{\frac{I_0 Z \alpha}{3(2\alpha + 3)}} \tag{3}$$

The strain fatigue was modified by means of the Morrow method. The Seegar algorithm was used to define the TC6 material properties [29]. The elastic modulus of TC6 was 123 GPa and the Poisson's ratio was 0.34. The material $\sigma_{HEL}$ was calculated with Poisson's ratio $\nu$ and dynamic yield strength of TC6.

## 3. Results and Discussion

### 3.1. Surface Roughness and Microhardness

The difference of surface roughness and microhardness between specimens is shown in Figure 5. Surface roughness was used to reflect the effect of LSP on surface flatness. The

results show that the surface roughness Ra increases as the plasticity increases. However, this relationship is not linear. Only the roughness of Ti-2 is less than 0.4 μm, yet the values of Ti-1 and Ti-3 are increased by 12.8% and 13.3%. In the engineering field, the flatness should be controlled in safe limits, because the rough surface would have a negative result in service life. By two impacts, the surface roughness of Ti-4 has increased to 0.491 μm, which is inappropriate for enhancing the surface. In fact, the altering of roughness consists of two periods although the LSP process is instantaneous. Firstly, the shock wave contacting with TC6 would cause local deformation, and roughness and heterogeneity rise in this process. Subsequently, the plastic deformation occurs, and the surface roughness decreases [30]. Lastly, the roughness is marginally increased.

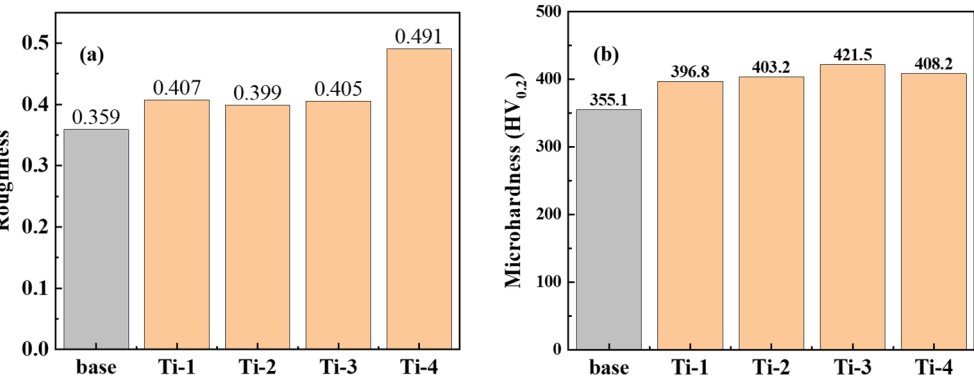

**Figure 5.** (**a**) Surface roughness, (**b**) surface microhardness.

Similarly, the surface microhardness has increased and is mainly attributed to plastic deformation as well as work hardening. As shown in Figure 5b, the microhardness of untreated sample is 355.1 HV. After LSP, the values improve to 396.8, 403.2, and 421.5 HV for Ti-1, Ti-2, and Ti-3 samples with one impact, respectively. The increased frequency is 11.7%–18.7%, compared to the untreated sample. However, after two impacts, the microhardness is only 408.2 HV at the same power density as Ti-2. This shows that the repeat procedure can no more refine the grains but destroy surface flatness. Because the first shock has induced more boundaries and tangled dislocations to hold back deformation, and the second power wave does not strengthen the surface anymore.

### 3.2. Microstuctures

In Figure 6a, the black regions represent hexagonal close-packed (hcp) $\alpha$ phases. Near the boundaries, the white regions are body-centered cubic (bcc) $\beta$ phases. Figure 6b–e indicate that, differing from the energy density, the thickness of the refined region is clearly different. The strengthening effect of Ti-2 and Ti-4 is better, for their thickness layers are thicker than Ti-1 and Ti-3. Hence, the power density of 3.69 GW/cm$^2$ was sufficient for TC6. However, there is no phase transformation in LSP process. Besides, the dislocations began to grow and move along the boundaries during the LSP procedure so as to achieve the action of surface strengthening. The surface strengthening is realized by plastic deformations while the surface is impacted by a high-pressure shock wave. The grains have been refined in the surface. As Figure 6f indicates, the LSP has changed the grains distribution [31], and the most profound effect was reflected in Ti-1 and Ti-2. Compared to the untreated specimen, the proportion of grain size $\alpha$ phases below 0.8 μm has reached 78% and 65%, respectively. For the $\beta$ phases, which were needle-shaped, the grain size could not be calculated. Under the condition of 6 J pulse energy, the fine refinement was about 40%. Similarly, the proportion after two impacts becomes 45% approximately, which is 44% lower than that of Ti-2. This result shows that the LSP has changed the microstructures of TC6. The distribution of the impact area gradually increases with the power density. Nevertheless, the increase of power density and impact number are close to saturation. The fine grains produced by plastic deformation will have a large

number of grain boundaries, which will hinder deformation and lead to work hardening. When the dislocations reached a certain value, the high-density dislocations can also cause work hardening like grain boundaries. The work hardening has prevented the surface strengthening from deforming like Ti-3 and Ti-4, as shown in Figure 6d,e. Overall, the microstructures of TC6 could be well controlled via reasonable regulation of LSP to improve the fatigue performance effectively.

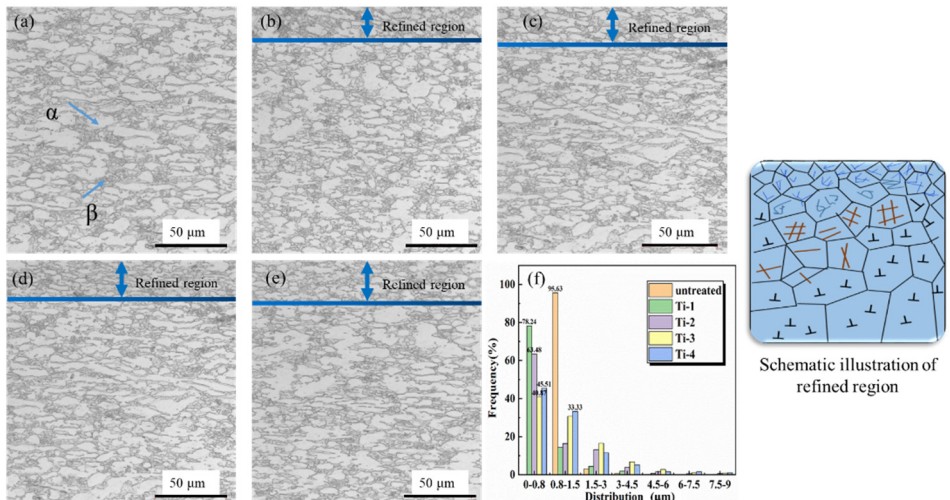

**Figure 6.** TC6 titanium alloy metallographic structure: (**a**) Untreated sample, (**b**) Ti-1, (**c**) Ti-2, (**d**) Ti-3, and (**e**) Ti-4, (**f**) statistical result of grain size distribution.

### 3.3. Residual Stress

Figure 7 shows a variation of residual stress along the distance from the surface. The measurements were proceeded for each 500 μm. Owing to machining, the residual stress of untreated sample is uniform −100 MPa. The negative symbol means compressive stress and the amplified compressive stress is beneficial to TC6 titanium alloy in defending against external force [32,33]. The LSP can induce high residual compressive stress along the depth direction. Besides, the residual compressive stress is derived from microstructure deformation. Owing to the plastic deformation, the microstructure was refined well, following stress concentration (compressive stress), according to the Hall–Petch relation [34]. The smaller the grain size, the higher the material strength. The surface residual stress is larger due to the stress wave in direct contact with the surface up to −600 MPa, as displayed in Figure 7.

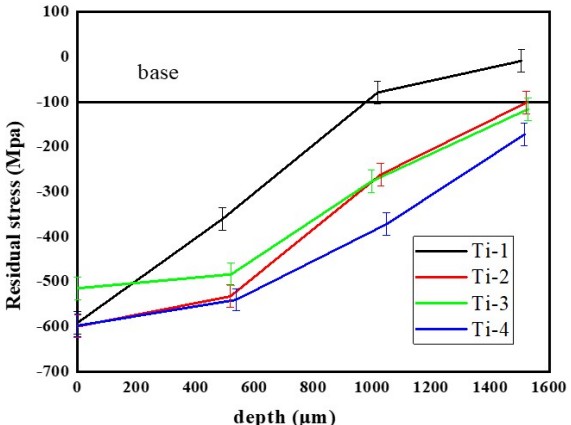

**Figure 7.** Residual stress of TC6 titanium alloy samples.

Along depth direction, the existence of compressive stress is limited owing to the presence of multiple grain boundaries and tangled dislocations [35]. The affected depth is about 1500 μm, indicating that the laser power along depth is to fade gradually. However, the Ti-1 sample's impact depth of less than 1000 μm contributed to the deficient energy power. As Figure 7 indicates, when the energy power becomes sufficient, the Ti-2 sample has a similar impact depth of 1500 μm with Ti-4. The two impact number makes a strengthening action of surface, which is attributed to the energy maintenance. Since the energy power exceeds the limit of TC6, the surface compressive stress of Ti-3 is less than that of Ti-2. In general, for these four samples, the values of compressive stress decrease gradually in a shallow layer [36]. The shock wave propagated in material represents energy decay. The energy is consumed by the material and then transformed into deformation. Moreover, the decay rate intensifies in the depth direction.

### 3.4. Failure Mechanism

Ti-2, as the typical LSP sample, was selected for the simulation and experiment in order to discuss and analyze the failure mechanism under the condition of continuously alternating load. The loading stress was from 500 to 580 MPa and the value of logarithmic of vertical axis was utilized as a standard to measure the fatigue life. For the horizontal axis, the scientific enumeration is more suitable to describe fatigue life. At the same stress level, when the service life is longer, the value of abscissa is larger, as shown in Figure 8. This shows that under relatively low stress, the strengthening effect is more obvious [37]. Figure 9a,b shows that the stress distribution conditions and stress concentration areas of the untreated model are marked by red. The three areas with the highest equivalent stress are located at the fillet of the variable cross-section, while the stress in the remaining areas is lower [38,39]. This high stress area will greatly reduce the service life of TC6. After LSP surface strengthening, the high-stress area is located at the rounded corner, as in the untreated sample [40]. However, this FEM model reliably shows that the life of the frame beam can be increased under the same loading stress. Figure 9a,b displays that the logarithmic fatigue life is 6.54 and 6.90 for the untreated sample and Ti-2. The fatigue life has increased by 29% under 518 MPa loading conditions. In experiments, under the loading conditions of 500 and 540 MPa, the fatigue life of the specimens was increased by 35% and 16%, respectively. The results are comparatively close which confirms that the complexity of the sample would not influence the strengthening effect. In a frame beam structure, the LSP can strengthen the fatigue performance the same as a sheet structure, owing to the surface microstructures and residual compressive stress [41,42].

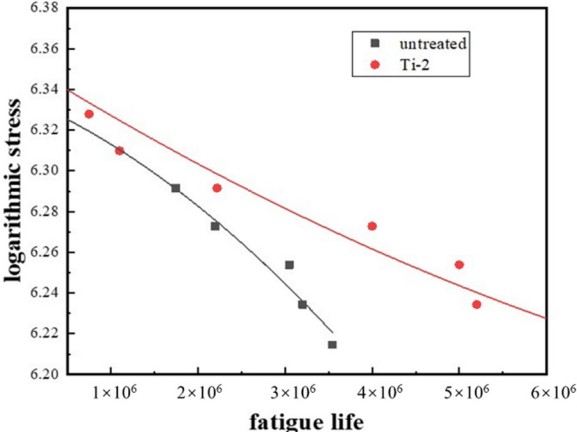

**Figure 8.** Fatigue life test results for untreated TC6 titanium alloy and Ti-2.

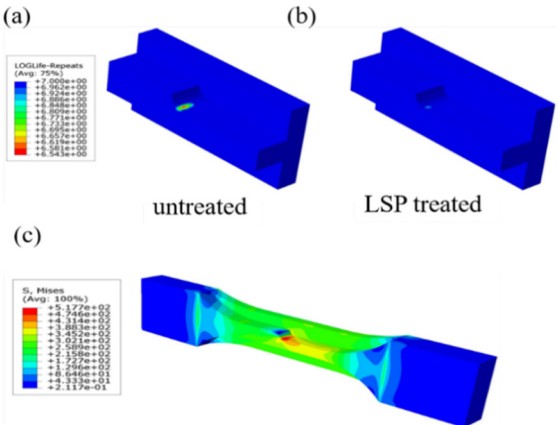

**Figure 9.** (**a**,**b**) Fatigue simulation of untreated and LSP-treated frame beams. (**c**) Static analysis results of frame beam structure simulation.

Furthermore, Figure 10 proves that the surface residual compressive stress can resist pressure and improve the fatigue performance significantly. The fracture morphologies of the original sample and Ti-2 are shown in Figure 10. Figure 10a expresses the fracture micro-morphology of the untreated sample. There is a giant region called crack growth inside the fracture [43,44]. If the crack growth region is amplified, the fatigue life of the specimen is shorter owing to the dimples. In virtue of LSP, after the initiation of crack, the residual compressive stress has dispersed the external force to prolong the service life in Figure 10c. Meanwhile, Figure 10d displays that the fatigue bands interweave with dimples to avoid breakage, and the microcracks propagate in all directions avoiding stress concentration. Nonetheless, the refined grains are beneficial for releasing the forces [45], and also the work hardening can further resist the impact of an external force to prevent fatigue damage.

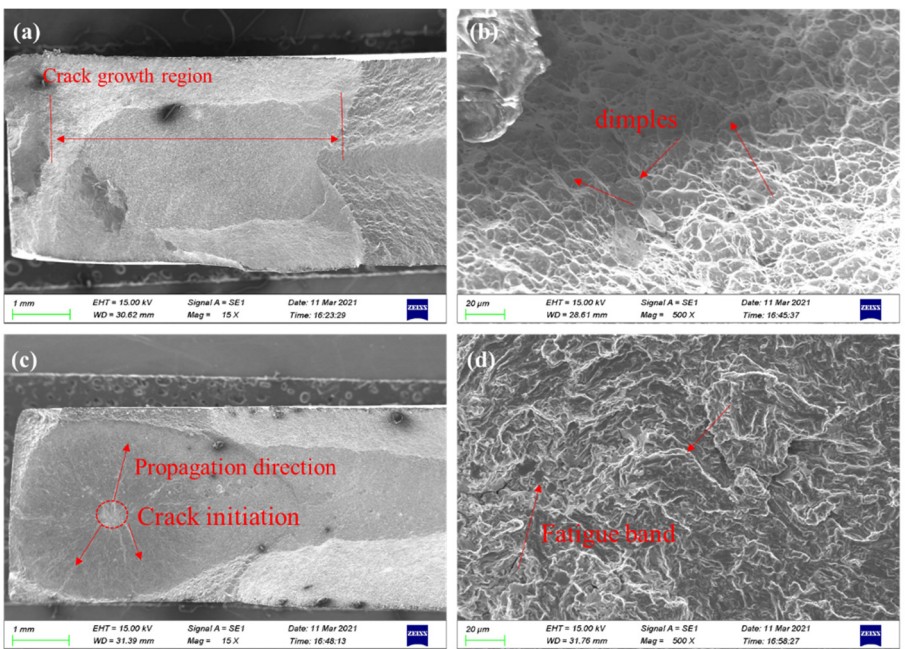

**Figure 10.** (**a**,**b**) Untreated TC6 titanium alloy and (**c**,**d**) LSP-treated Ti-2 samples.

## 4. Conclusions

In summary, TC6 titanium alloy was treated with different LSP processes, leading to a significant enhancement effect on the fatigue properties. Multiple laser shocks can

sharply enhance the material's performance. Besides, the number of laser shock differs in varied materials. The strengthening effect of Ti-2 was better than Ti-4 by two impacts at the same power density in experiments and FEM simulations, indicating that the power density of 4.71 GW/cm$^2$ by one impact was sufficient for TC6. For the titanium to be feasible for high energy density, where single energy source is sufficient, the repeating laser will weaken the strengthening effect instead. During the LSP procedure, the shock wave stroked the surface to induce plastic deformation, and consequently the grains were refined. The forces were kept inside as residual compressive during plastic deforming. The existence of residual compressive stress was able to improve fatigue performance and intensify the surface performance. Not only the microhardness improved without destroying the surface, but the propagation directions of microcracks changed. In the LSP process, a shock wave could significantly improve the surface residual stress and refine the surface grain to enhance the surface microstructure. The scale of $\alpha$ phases under 0.8 μm reached 78% from one impact. Owing to grain refinement and compressive stress, the microhardness of surface was up to 421.5 HV, which was 18.7% higher than an untreated specimen. By controlling the energy power and impacts, the surface residual stress could be enhanced to $-600$ MPa reaching 1500 μm, while the fatigue life was prolonged to 135%. By finite element simulation, the results of a beam frame structure were consistent with the experimental result of sheet specimens, which strongly corroborated the assertion that the LSP is able to strengthen the complex structure. This work reveals the relationship between the surface stress state and metal fatigue properties. By regulating the process parameters, desirable fatigue performance strengthening can be realized.

**Author Contributions:** Conceptualization, Y.S.; methodology, S.S.; validation, S.S., X.H., Z.C., Z.H., W.L.; writing—original draft preparation, S.S.; writing—review and editing, Y.S.; visualization, Z.C., Z.H.; funding acquisition, Y.S. All authors have read and agreed to the published version of the manuscript.

**Funding:** This work was supported by the National Natural Science Foundation of China (No. s52075246 and U1937206), the Project Funded by China Postdoctoral Science Foundation (No. 2019M661826), Open Fund of Key Laboratory of Icing and Anti/De-icing (No. IADL20190202, IADL20200407), the Project Funded by the Priority Academic Program Development of Jiangsu Higher Education Institutions.

**Institutional Review Board Statement:** Not applicable.

**Informed Consent Statement:** Not applicable.

**Data Availability Statement:** The data presented in this study are available within the article.

**Conflicts of Interest:** The authors declare no conflict of interest.

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
