# Peer review of "Understanding the Relations between Surface Stress State and Microstructure Feature for Enhancing the Fatigue Performance of TC6 Titanium Alloy"

_coatings, doi:10.3390/coatings11101261_

Round 1
Reviewer 1 Report
Dear authors,
Thank you for your very interesting paper on laser shock peening of TC6 titanium.
In general, the paper offers interesting results and the process parameter variation is useful for practical applications. The following comments are given and should preferably be included in the revised version of the paper:
- General: there are a number of typos and the grammar makes it sometimes difficult to understand sentences. Please have a native speaker proofread the manuscript before resubmission.
- Line 32-33: The meaning of the sentence is unclear. Please revise the sentence.
- Line 65: It is not true that this has not been investigated so far. A quick search revealed for example the following paper: not true:
Luo, X., Dang, N., & Wang, X. (2021). The effect of laser shock peening, shot peening and their combination on the microstructure and fatigue properties of Ti-6Al-4V titanium alloy. International Journal of Fatigue, 153, 106465
- Line 110: it is assumed you refer to “grain morphology”. Is that correct?
- Line 118: please add the test frequency and did you use a cooling device to avoid specimen heating?
- Line 152: Please add a reference to Brown-Miller critical plane algorithm.
- Line 164: which typo of roughness definition is used (Ra, Rz)?
- Figure 6: Please add higher resolution figures.
- Line 260: the sentence is colloquial, please improve.
- Figure 8: It is impossible to read legend. Where does the localized high stress comes from? Were the specimen loaded in bending and why is there a difference in thickness?
- Figure 9: fatigue tests are typically displayed in loglog diagrams. Please add the logarithm of stress. Furthermore, please change the axes to 10^6 scientific number format. Finally, why was the Ti-2 process chosen for testing and not the others?
- Conclusions: can you give recommendations for process parameters and add a discussion based on other studies?
- Finally, my main concern with your study is the lack of a general discussion of your results. Please discuss your method and the results based on the presented state-of-the-art.
Author Response
Response to Reviewer
Thank you very much for your positive evaluations and the valuable suggestions on our manuscript. We have revised carefully the paper and addressed all these comments. Also, the modifications and supplements were clearly highlighted using yellow in revised manuscript. The main corrections and the response to reviewer’s comments are listed as follows:
I look forward to hearing from you soon.
Best regards!
Assoc. Prof. Yizhou Shen
Nanjing University of Aeronautics and Astronautics, P. R. China
The authors present an interesting paper on laser shock peening of TC6 titanium. In general, the paper offers interesting results and the process parameter variation is useful for practical applications. General: there are a number of typos and the grammar makes it sometimes difficult to understand sentences. Please have a native speaker proofread the manuscript before resubmission.
- Line 32-33: The meaning of the sentence is unclear. Please revise the sentence.
Author reply: Thank you for pointing out this problem in our manuscript! According to your suggestion, we have rewritten the sentence to clearly show the advantages of TC6.
- Line 65: It is not true that this has not been investigated so far. A quick search revealed for example the following paper: not true:
Luo, X., Dang, N., & Wang, X. (2021). The effect of laser shock peening, shot peening and their combination on the microstructure and fatigue properties of Ti-6Al-4V titanium alloy. International Journal of Fatigue, 153, 106465.
Author reply: We gratefully appreciate for your valuable suggestion. We have carefully read the reference you provided and cited it as references. Combing our experimental studies and this reference, the relationship between microstructure and fatigue property was supplemented and discussed.
- Line 110: it is assumed you refer to “grain morphology”. Is that correct?
Author reply: Thank you so much for your careful check! We have added the “grain morphology” to the revised manuscript at Line 110.
- Line 118: please add the test frequency and did you use a cooling device to avoid specimen heating?
Author reply: Thank you for the above suggestion! At line 118, we have provided detailed descriptions of high-frequency fatigue teseting machine. The test frequency was 150 Hz, and the device was equipped with radiator to avoid heating.
- Line 152: Please add a reference to Brown-Miller critical plane algorithm.
Author reply: Thanks very much! We have made a supplement reference [28] according to your comments.
- Line 164: which typo of roughness definition is used (Ra, Rz)?
Author reply: Thank you for your valuable reminder! We have used Ra as evaluation paramenter, and added specific detail (at line 177).
- Figure 6: Please add higher resolution figures.
Author reply: We are appreciative of your suggestion! According to your suggestion, we have replaced the figures in higher resolution.
- Line 260: the sentence is colloquial, please improve.
Author reply: Thank you for carefully reminding! We have corrected this sentence in our revised manuscript (at line 273).
- Figure 8: It is impossible to read legend. Where does the localized high stress comes from? Were the specimen loaded in bending and why is there a difference in thickness?
Author reply: Thanks very much! The local high stress is mainly due to the color transition in the simulation in Figure 8. As can be seen from the axis on the left of the image, different colors meant different stress states, so the stress was more concentrated in the area with brighter colors (at line 255-258). At the same time, the difference in thickness was due to the fact that the simulation adopted the graph with rounded corners, whose shape was more complex than that of the plate(in line 258-260). There were difficulties in the actual sample, so the simulation was carried out.
- Figure 9: fatigue tests are typically displayed in log diagrams. Please add the logarithm of stress. Furthermore, please change the axes to 10^6 scientific number format. Finally, why was the Ti-2 process chosen for testing and not the others?
Author reply: Thanks very much! According to your requirements, we have modified Figure 9 and changed the axes to 106 scientific number format. For Ti-2 specimen was tested with more smooth surface, refined grains as well as residual compressive stress. Its comprehensive performance is better and can effectively enhancing fatigue performance. Therefore, the set of parameters of Ti-2 were more appropriate to be selected for testing.
- Conclusions: can you give recommendations for process parameters and add a discussion based on other studies?
Author reply: Thank you for the above suggestion! In conclusion, we have provided process parameters and discussion based on other studies (at line 291-293).
- Finally, my main concern with your study is the lack of a general discussion of your results. Please discuss your method and the results based on the presented state-of-the-art.
Author reply: We totally understand your concern. In conclusion section, we have provided more general discussion as well as results based on presented state-of-the-art (at line 300-302 and line 305-307).

Reviewer 2 Report
In the actual form, the manuscript is not suitable for publication.
First of all, the authors performed numerical simulation. But what is the point? The reader cannot get any information from the results presented in Figure 8. If the authors wanted to present a relevant numerical simulation that would be about the simulation of the laser treatment in order to estimate the value of the residual stresses.
In addition, the reviewer suggests the following improvement.
Introduction :
Page 2, lines 58-60 ‘Therefore, 58
LSP can significantly improve the fatigue resistance, wear resistance, and stress corrosion resistance of metal materials [17,18].’ You are speaking about room temperature fatigue or fatigue at cryogenic temperature, isn’t it.
Experiments and simulations
Page 2, lines 74-75 : Were the plates in which you prepare the specimens prepared by rolling? Did you study crystallographic texture? Were the specimens machined along the rolling direction?
Page 4, lines 111-113: In the calculation of the residual stresses, did you take into account of the relief of the stress owing to the removal of the upper layers by electropolishing technique? See Moore, M. and Evans, W., "Mathematical Correction for Stress in Removed Layers in X-Ray Diffraction Residual Stress Analysis," SAE Technical Paper 580035, p6, 1958.
Page 4, lines 117-118: ‘Titanium alloy is sensitive to stress concentration [24,25], and we pasted electrical sheet in the holding section, as shown in Figure 3.’ Sorry, what is the role of the electrical sheet? Did you use an electrical method to detect crack and to follow the crack propagation with the number of cycle during fatigue tests?
Page 4, lines 137-138: ‘ABAQUS/Explicit have been utilized to analyze the shock propagation and stress state. In these models, the cases are solved in different linear and non-linear equations.’ Please give details about these equations?
Results and discussion
Page 5, lines 173-174: ‘Similarly, the surface microhardness has increased and is mainly attributed to plastic deformation as well as work hardening.’ The authors did not observe a grain refinement?
Page 5, figure 5: if the surface roughness has an important effect on the crack initiation, why the author did not polish the surfaces?
Page 6, lines 185-186 and Figures 6: These figures do not bring any information, except that the grains are slightly elongated which would suggest a slight crystallographic texture. Could the authors present images showing the surface of the materials after laser surface? Remarque: the volume fraction of the beta phase seems to be relatively high for this TC6 alloy. If you look at the microstructure at a finer scale, you might observe fine alpha phase in the beta phase.
Page 6, line 210-211: ‘The measurements were proceeded for each 100 μm.’ Why so few data points in Figure 7?
Figure 9: The shape of the stress-number of cycle to failure is strange. I do not know which textbook(s) you have read but the Wolher curves usually do not follow such representation. Consequently, you have difficulty fitting with a straight line.
English :
Page 1, line 31-33 : Not only with the advantages of high specific strength and good corrosion resistance, but this alloy also has the good plasticity to process into any shapes and high-temperature (at 400–450 °C) work capacity. ‘the good plasticity’? Please rewrite in proper English
Page 1, line 42-44 : Shot peening method requires simple equipment, yet the reinforcement layer is relatively thin with smaller depth [5-7]. Please rewrite this sentence particularly ‘reinforcement’ which is certainly not the proper qualifying adjective and ‘relatively thin with smaller depth’, which is not very clear.
Page 3, line 98 : E = 123 MPa: no! GPa
Page 8, lines 266: ‘Still, the refined grains are beneficial to diffusion of the forces [45, 46] ...’ Diffusion is not the proper term.
Author Response

(The authors gave the same response as above.)

Round 2
Reviewer 1 Report
Thank you for making the requested changes to the manuscript. The paper has largely improved. However, there are still plenty of typos and minor grammatical mistakes that should be corrected. Please have a native speaker check the manuscript or use the MDPI language editing service.
Author Response
Response to Reviewer
Thank you very much for your suggestions on our manuscript. We have revised carefully the paper. Also, the modifications and supplements were clearly highlighted using yellow in revised manuscript.
I look forward to hearing from you soon.
Best regards!
Assoc. Prof. Yizhou Shen
Nanjing University of Aeronautics and Astronautics, P. R. China
Thank you for making the requested changes to the manuscript. The paper has largely improved. However, there are still plenty of typos and minor grammatical mistakes that should be corrected. Please have a native speaker check the manuscript or use the MDPI language editing service.
Author reply: Thank you for pointing out problem in our manuscript! According to your suggestion, we have updated the description with less possibility of misunderstanding. Finally, a native speaking professor in our university was invited to further refine the English writing.

Reviewer 2 Report
Improvements are still required before publication.
As already expressed in the first review, the quality of the figures 6 are not satisfying. These figures should show the microstructure beneath the surface.
Also but when you are dealing with a 2 phase material, information about both phases are necessary.
Introduction
Line 42 : Shot peening (SP) method requires simple equipment, yet the strengthened layer is relatively thin with small depth below 0.8 mm [5-7]. The reference 6 deals with laser penning, not shot penning!
Line 44 : Another surface strengthening method (i.e., low-plasticity rolling) can obtain a deeper strengthened layer and lower surface roughness [8-10]. The reference 10 deals with laser penning, not rolling!
Did you hear about ultrasonic surface mechanical attrition treatment (SMAT)
Line 118 : Titanium alloy is sensitive to stress concentration [24,25], and we pasted electrical sheet in the holding section, as shown in Figure 3. Unfortunately, the references are not appropriated: in reference 24 the material is Ni-Ti and in reference 25, le material is a beta Ti alloy, not an alpha-beta Ti alloy.
Experiments and simulations
Line 140 : why the letter t is found in the sentence: ‘The dynamic equation t can be expressed as follows’?
Line 141 : Please explicit the symbol omega and tau as you did for the other variables in eq.2.
Lines 147-157: As the simulation of the laser shock penning by ABAQUS been described in a publication, the manuscript of a PhD thesis, or elsewhere? Further details for this simulation would be interesting for the reader.
Results and discussion
Microstructure : We could expect a significant variation of the microstructure beneath the surface, while Figure 6 does not reveal much changes. That’s surprising. Indeed we should expect a gradual change of the microstructure from the surface towards the core of the sample…
Residual stress
Lines 230-231 : Owing to the plastic deformation, the microstructure has been refined, following stress concentration (compressive stress) in good way. What does that means? Did the authors try to apply the Hall-Petch relation?
Lines 234-235: Along depth direction, the existence of compressive stress is limited owing to the presence of multiple grain boundaries and tangled dislocations. Did the authors observed dislocations? And how? Did the authors performed analyses in MEB equipped with EBSD technique or did they observe the microstructure by TEM?
Figure 7 : no scatter band… What is the uncertainties in the determination of the value of the residual stress? In our laboratory such measures are accurate at about plus or minus 25 MPa.
Line 251: the authors wrote : The loading stress was 500 MPa, 540 MPa, and 580 MPa … however the data in figure 9 displayed more than 3 different stress levels.
Lines 251-252 : … and the value of logarithmic was utilized as a standard to measure the fatigue life.’ Also line 262-263 : ‘As Figure 8(a,b) displays that the logarithmic fatigue life is 6.54 and 6.90 for untreated sample and Ti-2.’ Why presenting the value of the stress level expressed in logarithm? Please present the vertical axis as you did for the horizontal axis.
Conclusion:
Line 289 : Multiple laser shock can sharply enhance the material performance. Indeed only the results for a single shock are presented!
Line 300 : The scale of grains size under 0.8 μm has reached 78% by one impact. The grain size 0.8 is for the alpha phase, isn’t it. Okay, but when you are dealing with a 2 phase material, information about both phases are necessary. The locations of the second is important, the morphology, etc…
English:
Line 163 : sentence : Where sigma is the dynamic response yield strength at high strain rate (MPa) and λ and α are Raman constants related to the material's elastic modulus E and Poisson's ratio. Where?
Lines 202-203: It occurred the grains refinement in the surface. What does that mean?
Author Response
Response to Reviewer
Thank you very much for your valuable suggestions on our manuscript. We have revised carefully the paper and addressed all these comments. Also, the modifications and supplements were clearly highlighted using yellow in revised manuscript. The main corrections and the response to reviewer’s comments are listed as follows:
I look forward to hearing from you soon.
Best regards!
Assoc. Prof. Yizhou Shen
Nanjing University of Aeronautics and Astronautics, P. R. China
Improvements are still required before publication. As already expressed in the first review, the quality of the figures 6 are not satisfying. These figures should show the microstructure beneath the surface. Also but when you are dealing with a 2 phase material, information about both phases are necessary.
Author reply: Thank you so much for your positive comments and careful check. We have revised the syntactically incorrect statement accordingly. Also, we have improved the inappropriate parts referring to your advice.
- Line 42 : Shot peening (SP) method requires simple equipment, yet the strengthened layer is relatively thin with small depth below 0.8 mm [5-7]. The reference 6 deals with laser penning, not shot penning!
Author reply: Thank you so much for your careful check! According to your suggestion, we have cited more appropriate references to make sure the uniformity of article and references.
- Line 44 : Another surface strengthening method (e., low-plasticity rolling) can obtain a deeper strengthened layer and lower surface roughness [8-10]. The reference 10 deals with laser penning, not rolling!
Author reply: Thanks for your kindly reminding! According to your suggestion, we have cited more appropriate references.
- Did you hear about ultrasonic surface mechanical attrition treatment (SMAT).
Author reply: The power of ultrasonic surface mechanical is lower than that of laser. In most cases, the ultrasonic surface mechanical attrition treatment is used for machinable materials including carbon steel, aluminum alloy, magnesium alloy and other materials. For titanium alloy with high strength, SMAT was not suitable as a surface strengthening method to be introduced.
- Line 118 : Titanium alloy is sensitive to stress concentration [24,25], and we pasted electrical sheet in the holding section, as shown in Figure 3. Unfortunately, the references are not appropriated: in reference 24 the material is Ni-Ti and in reference 25, le material is a beta Ti alloy, not an alpha-beta Ti alloy.
Author reply: Accordingly, we have cited more appropriate references to make sure the material properties in the references are consistent in this study.
- Line 140 : why the letter t is found in the sentence: ‘The dynamic equation t can be expressed as follows’?
Author reply: Thank you for pointing out this problem in our manuscript! We have revised the mistake in the manuscript at line 140.
- Line 141 : Please explicit the symbol omega and tau as you did for the other variables in eq.2.
Author reply: According to your suggestion, we have added the introduction section of symbol where Ω means the global scope and τ means bounds at lines 141.
- Lines 147-157: As the simulation of the laser shock penning by ABAQUS been described in a publication, the manuscript of a PhD thesis, or elsewhere? Further details for this simulation would be interesting for the reader.
Author reply: Thank you so much for your reminding! We have added more details at section 2.4 and highlighted using yellow in revised manuscript.
- Microstructure : We could expect a significant variation of the microstructure beneath the surface, while Figure 6 does not reveal much changes. That’s surprising. Indeed we should expect a gradual change of the microstructure from the surface towards the core of the sample…
Author reply: In most cases, a gradual change of the microstructure from the surface towards the core can reveal the strengthening effect of laser shock peening. In fact, the aim of this article was to understand the relations between surface stress state and microstructure feature, however, we can hardly detect the microstructure changes of titanium alloy from the side.
- Lines 230-231 : Owing to the plastic deformation, the microstructure has been refined, following stress concentration (compressive stress) in good way. What does that means? Did the authors try to apply the Hall-Petch relation?
Author reply: Thank you so much for your reminding! According to your suggestion, we have added the Hall-Petch to further explain the relationship between microstructure and stress concentration.
- Lines 234-235: Along depth direction, the existence of compressive stress is limited owing to the presence of multiple grain boundaries and tangled dislocations. Did the authors observed dislocations? And how? Did the authors performed analyses in MEB equipped with EBSD technique or did they observe the microstructure by TEM?
Author reply: Thank you so much for your careful check! In fact, we did not observe the presence of multiple grain boundaries and tangled dislocations without TEM or EBSD. However, this conclusion was mainly based on the results of previous papers, and we have added relevant references in the approporiate place.
- Figure 7 : no scatter band… What is the uncertainties in the determination of the value of the residual stress? In our laboratory such measures are accurate at about plus or minus 25 MPa.
Author reply: We have added the scatter band in the Figure.7 as your advice.
- Line 251: the authors wrote : The loading stress was 500 MPa, 540 MPa, and 580 MPa … however the data in figure 9 displayed more than 3 different stress levels.
Author reply: Thank you so much for your careful check! The measurements were proceeded from 500MPa to 580 MPa, and we have fixed this error.
- Lines 251-252 : … and the value of logarithmic was utilized as a standard to measure the fatigue life.’ Also line 262-263 : ‘As Figure 8(a,b) displays that the logarithmic fatigue life is 6.54 and 6.90 for untreated sample and Ti-2.’ Why presenting the value of the stress level expressed in logarithm? Please present the vertical axis as you did for the horizontal axis.
Author reply: Thank you so much for your careful check! Logarithm was used for horizontal while scientific counting method is used in vertical axis. In this way of expression is mainly to more intuitive to see the trend of change and strengthening effects. The expression of horizontal and vertical coordinates in this way is mainly based on the S-N curve and papers of others. Meanwhile, we have revised the manuiscraipt in more exact words avoiding ambiguity.
- Line 289 : Multiple laser shock can sharply enhance the material performance. Indeed only the results for a single shock are presented!
Author reply: Thank you so much for your careful check! Referring to previous papers and reviews, we have added some general conclusions, that is, multiple laser shocks can sharply enhance the material performance. However, the times of laser shock is different in different materials. In this article, the comprehensive performance of single shock was better. The reason for this phenomenon is that titanium alloys can be subjected to high energy shocks. In situations where a single energy is sufficient, a second shock will weaken the effect instead.
- Line 300 : The scale of grains size under 0.8 μm has reached 78% by one impact. The grain size 0.8 is for the alpha phase, isn’t it. Okay, but when you are dealing with a 2 phase material, information about both phases are necessary. The locations of the second is important, the morphology, etc…
Author reply: For two phase material, the locations and the scale of grain sizes about both phases are necessary. However, in this study, the current tests have not been able to clearly observe the boundary and distribution between the two phases. In addition, observing the more precise distribution is able to study the relationship between phase and the strengthening effects. It can be a deeper research. For this article, the rough disturibution of grains can be helpful to understand the relationship between microstructure.
- Line 163 : sentence : Where sigma is the dynamic response yield strength at high strain rate (MPa) and λ and α are Raman constants related to the material's elastic modulus E and Poisson's ratio. Where?
Author reply: Thank you so much for your careful check! According to your reminding, we have adjusted the section of lines 163 to eq.1 at line 98, the right place.
- Lines 202-203: It occurred the grains refinement in the surface. What does that mean?
Author reply: Thanks for your kindly reminding! According to your suggestion, we have adjusted this word in the revised manuiscript.

Round 3
Reviewer 2 Report
As writing in the previous review:
Improvements are still required before publication.
As already expressed in the first review, the quality of the figures 6 are not satisfying. These figures should show the microstructure beneath the surface.
Also but when you are dealing with a 2 phase material, information about both phases are necessary.
Shot penning is an old fashion technique. When doing the literature survey you never read an article about ultrasonic surface mechanical attrition treatment (SMAT) : please look at the article : https://www.nature.com/articles/s41524-019-0171-6
Results and discussion:
Indeed the fatigue life is improved only for low stress values!
Conclusion:
Line 289 : Multiple laser shock can sharply enhance the material performance. Indeed only the results for a single shock are presented!
Author Response
Response to Reviewer
Thank you very much for your valuable suggestions on our manuscript. We have revised carefully the paper and addressed all these comments. Also, the modifications and supplements were clearly highlighted using yellow in revised manuscript. The main corrections and the response to reviewer’s comments are listed as follows:
I look forward to hearing from you soon.
Best regards!
Assoc. Prof. Yizhou Shen
Nanjing University of Aeronautics and Astronautics, P. R. China
Improvements are still required before publication. As already expressed in the first review, the quality of the figures 6 are not satisfying. These figures should show the microstructure beneath the surface. Also but when you are dealing with a 2 phase material, information about both phases are necessary.
Author reply: Thank you so much for your positive comments and careful check. We have revised the figure 6 to show the microstructure beneath the surface accordingly. Also, we have added the detailed information about both phases.
- Shot penning is an old fashion technique. When doing the literature survey you never read an article about ultrasonic surface mechanical attrition treatment (SMAT) : please look at the article : https://www.nature.com/articles/s41524-019-0171-6
Author reply: I have modified the introduction section of the manuiscript according to your recommending and relevant literatures.
- Indeed the fatigue life is improved only for low stress values!
Author reply: Thank you so much for your correcting. We have revised the incorrect statement referring to your advice.
- Line 289 : Multiple laser shock can sharply enhance the material performance. Indeed only the results for a single shock are presented!
Author reply: Sorry, because of our negligence, we did not explain the conclusion in the article before. This time we have added the corresponding explanation in the conclusion section in the manuscript.
